# Unsupervised Learning of Causal Relationships from Unstructured Data

## Abstract

Endowing deep neural networks with the ability to reason about cause and effect would be an important step to make them more robust and interpretable. In this work we propose a variational framework that allows deep networks to learn latent variables and their causal relationships from unstructured data, with no supervision, or labeled interventions. Starting from an abstract Structural Equation Model (SEM), we show that maximizing its posterior probability yields a similar construction to a Variational Auto-Encoder (VAE), but with a structured prior coupled by non-linear equations. This prior represents an interpretable SEM with learnable parameters (such as a physical model or dependence structure), which can be fitted to data while simultaneously learning the latent variables. Unfortunately, computing KL-divergences with this non-linear prior is intractable. We show how linearizing arbitrary SEMs via back-propagation produces local non-isotropic Gaussian priors, for which the KL-divergences can be computed efficiently and differentiably. We propose two versions, one for IID data (such as images) which detects related causal variables within a sample, and one for non-IID data (such as video) which detects variables that are also related over time. Our proposal is complementary to causal discovery techniques, which assume given variables, and instead discovers both variables and their causal relationships. We experiment with recovering causal models from images, and learning temporal relations based on the Super Mario Bros videogame.

## 1 Introduction

Human reasoning and decision-making is often underpinned by cause and effect: we take actions to achieve a desired effect, or reason that events would have happened differently had we acted a certain way – or if conditions had been different. Similarly, scientific inquiry uses the same tools, albeit more formalized, to build knowledge about the world and how our society can affect it (Popper, 1962). When building algorithms that automatically build statistical models of the world, as is common in machine learning practice, it would then be desirable to imbue them with similar inductive priors about cause and effect (Glymour et al., 2016). In addition to being more robust than statistical models which only characterize the observational distribution (Peters et al., 2017), they would allow reasoning about changing conditions outside the observed distribution (e.g. counterfactual reasoning). They would also allow communicating their inner workings more effectively – allowing us to ask "why" a given conclusion was reached, much in the same way that we do in scientific communication.

Despite still being actively researched, there is now a mature body of work on understanding whether two or more variables are related as cause and effect (Peters et al., 2017). Many techniques assume that the variables are given, and concern themselves with finding relationship between them (Spirtes & Glymour, 1991; Chickering, 2003; Lorch et al., 2021). On the other hand, an advantage of modern deep neural networks is that they learn intermediate representations that do not have to be manually labeled (Yosinski et al., 2015), and effective models can be trained without supervision (Kingma & Welling, 2014). An important question then arises: can a deep network simultaneously discover latent variables in the data and establish cause-effect relationships between them?

We focus on learning Additive Noise Models (ANM) with Gaussian noise, which are identifiable (i.e. causal directions are distinguishable) as long as the functions relating the variables of interest are *not* linear (Hoyer et al., 2008). This model fits well a variational learning framework, and so we are able

to derive an analogue of a Variational Auto-Encoder (VAE) (Kingma & Welling, 2014) where the prior, rather than being an uninformative Gaussian, corresponds exactly to the ANM. When the ANM is linear with Gaussian noise, the joint probability of the variables also becomes Gaussian, and it is easy to perform variational inference. The dependencies between variables will then be expressed in the covariance matrix's sparsity structure. However, as mentioned earlier to make the causal directions identifiable the model *cannot* be linear (Hoyer et al., 2008). We resolve this difficulty by learning models that are locally linear, but globally non-linear. This approach affords the full generality of a non-linear ANM, with the simplicity of variational inference on Gaussian models.

In summary, our contributions are:

- A rigorous derivation of the variational Evidence Lower Bound (ELBO) of an Additive Noise Model (ANM), allowing efficient inference of Structural Equation Models (SEM) with deep networks.
- A linearization method leveraging automatic differentiation to construct a local Gaussian approximation of arbitrary non-linear ANMs.
- A temporally-aware specialization of the causal ANM that encodes causal directions implicit in the arrow-of-time and is suitable for high-dimensional time series data such as video.
- Experiments demonstrating that the proposed method is able to fit latent variables with a dependence structure in high-dimensional data, namely a synthetic image dataset and video game based data.

## 2 RELATED WORK

Our work lies on the intersection of causality, variational inference, representation learning, and high-dimensional unstructured input domains.

Causal inference deals with determining the causes and effects from data. Causal discovery methods generally focus on recovering the causal graph responsible for generating the observed data, e.g. Spirtes & Glymour (1991); Chickering (2003) (for an overview of methods in see Peters et al. (2017)). However, these methods are largely applied to structured datasets such as medical (Brooks-Gunn et al., 1992; Sachs et al., 2005; Louizos et al., 2017) or economics data LaLonde (1986) where the observed variables are provided by domain specialists. In contrast, we focus on unstructured data where the variables are not provided *a priori*.

Variational inference is a way of performing inference by solving an optimisation problem. A popular instance is the Variational Auto-Encoder (VAE) (Kingma & Welling, 2014) which aims to extract a useful latent representation of the data by encoding and decoding it back. Traditionally the VAE prior is assumed to be an isotropic Gaussian distribution and the aim is to extract independent latent variables such as in the $\beta$-VAE (Higgins et al., 2016b) and FactorVAE (Kim & Mnih, 2018). There are works which use hierarchical priors such as iteratively conditioning each variable on its preceding variable in the Ladder-VAE (Sønderby et al., 2016) and conditioning each variable on all its predecessors in NVAE (Vahdat & Kautz, 2020) and VDVAE (Child, 2021). We also use a prior conditioning each variable on its predecessors but this comes as a natural consequence of basing our prior on a structural equation model (SEM).

Recently there has been a growing interest in representation learning based on causal principles. For instance, the CausalVAE (Yang et al., 2021) learns independent latent variables which are then composed to form causal relationships, however they only consider linear relationships between variables. Other works use different approaches to VAEs for causal learning such as the CausalGAN (Kocaoglu et al., 2018) which use generative adversarial networks. Yet another line of work focuses on modelling object dynamics from video such as Li et al. (2020), however they use specialised modules for detecting keypoints and future prediction. Another line of work uses graph neural networks to infer an interaction graph such as Kipf et al. (2018); Löwe et al. (2022), but they do not deal with image or video data. Lippe et al. (2022) focus on causal learning with the knowledge of interventions, whereas we assume no such knowledge. Another line of work such as Lachapelle et al. (2022) and the iVAE (Khemakhem et al., 2020) uses non-linear Independent Component Analysis theory. Locatello et al. (2020) explore using a small number of labeled examples for learning. Walker et al. (2021) use a VQ-VAE for video future prediction using a hierarchical prior but do not focus on causal relationships.

## 3 BACKGROUND

In this section, we will give a self-contained overview of several results from variational and causal inference that we build upon. While they are not new, bringing them together under one formulation offers new insights and challenges, which we solve in sec. 4.

Our goal is to fit a distribution $p(x)$, defined over an input space $x \in \mathbb{R}^m$, to an empirical distribution $\hat{p}(x)$ composed of finite samples, by choosing the optimal $p \in \mathcal{P}$ out of a set of candidate distributions $\mathcal{P}$ (e.g. parameterized by a neural network). We do this by minimizing their KL-divergence $D_x$ computed over $x$, or equivalently maximizing the expected log-likelihood of $p$ over the dataset $\hat{p}$:

$$p^* = \arg\min_{p \in \mathcal{P}} D_x(\hat{p}(x)||p(x)) = \arg\max_{p \in \mathcal{P}} \mathbb{E}_{x \sim \hat{p}(x)}[\ln p(x)]. \tag{1}$$

We now introduce a set of latent variables $z \in \mathbb{R}^d$, which we can marginalize to compute

$$p(x) = \int p(z)p(x|z)\mathrm{d}z, \tag{2}$$

in terms of a conditional distribution $p(x|z)$ and a "prior" distribution over latents $p(z)$. In a standard VAE, this prior is an isotropic Gaussian distribution (Kingma & Welling, 2014), while other structured priors are possible (Sønderby et al., 2016; Vahdat & Kautz, 2020; Tomczak & Welling, 2018). In this work, however, we will define it as a Structural Equation Model (SEM) (Pearl, 2009) (section 3.2).

### 3.1 VARIATIONAL INFERENCE

We can now apply standard tools of variational inference (Bishop, 2006, Ch. 10) to eq. 1 and replace the intractable marginalization (eq. 2) with an optimization of $q \in \mathcal{Q}$ over a variational family of distributions $\mathcal{Q}$ (in essence, training an additional neural network $q$). Eq. 1, when marginalized (eq. 2) is equivalent to (Kingma & Welling, 2014):

$$p^* = \arg\max_{p \in \mathcal{P},\, q \in \mathcal{Q}} \mathbb{E}_{x \sim \hat{p}(x)} \left[ \mathbb{E}_{z \sim q(z|x)} [\ln p(x|z)] - D_z \left( q(z|x)||p(z) \right) \right]. \tag{3}$$

The first term in eq. 3 amounts to a reconstruction error, and the second term matches the latent variables to the prior. In practice, $q(z|x)$ and $p(x|z)$ in eq. 3 are often defined as Gaussian distributions parameterized by neural networks. These are the functions $\mu_q(x) \in \mathbb{R}^d$ and $\Sigma_q(x) \in \mathbb{R}^{d \times d}$ for the encoder $q$, and $\mu_p(z) \in \mathbb{R}^d$ and $\sigma_p(z) \in \mathbb{R}^d$ for the decoder $p$, which parameterize the means and covariances of the distributions:

$$q(z|x) = \mathcal{N}(z|\mu_q(x), \Sigma_q(x)), \qquad p(x|z) = \mathcal{N}(x|\mu_p(z), \mathrm{diag}(\sigma_p(z))). \tag{4}$$

The KL-divergence between prior $p(z)$ and variational posterior $q(z|x)$ can be computed in closed form when both are Gaussians. Note that the encoder usually outputs a diagonal covariance matrix (motivated by the fact that an isotropic prior is also diagonal), but here we allow it to output full covariances $\Sigma_q(x)$, which will be important later (sec. 4).

### 3.2 STRUCTURAL EQUATION MODEL

We now consider a set of variables $y$, which have a dependency structure defined by a directed acyclic graph (DAG) $G$ (i.e. there is an edge $(i \to j) \in G$ if $y_i$ is required to compute $y_j$). Additionally, the generation process for each variable $y_i$ can be described by a sequence of non-linear equations:

$$y_i = f_i(y_{\mathrm{pa}_G(i)}) + n_i, \tag{5}$$

where $f_i : \mathbb{R}^{|\mathrm{pa}_G(i)|} \mapsto \mathbb{R}^1$ is an arbitrary deterministic function, $\mathrm{pa}_G(i)$ denotes the indices of parent nodes of $i$ in $G$, and $n_i \sim \mathcal{N}(n_i|0, \sigma_i^2)$ is an independent zero-mean noise variable, assumed to be Gaussian. This represents a Structural Equation Model (SEM), more specifically an Additive Noise Model (ANM) (Peters et al., 2017, Ch. 4.1.4) with Gaussian noise, which has the joint probability:

$$p(y) = \prod_{i=1}^{d} p_i(y_i \,|\, y_{\mathrm{pa}(i)}) = \prod_{i=1}^{d} \mathcal{N}(y_i - f_i(y_{\mathrm{pa}(i)}) \,|\, 0, \sigma_i^2). \tag{6}$$

Practical methods for causal learning with ANMs typically assume the variables $y$ are observed, and are concerned with recovering the true causal graph $G$ (and sometimes the functions $f$) that generated the data (Hoyer et al., 2008; Peters et al., 2017).

### 3.2.1 SCORE-BASED MODEL SELECTION

The formulation so far has assumed that the causal graph $G$, which encodes all the dependencies between variables, is given. We can, however, make use of a result by Nowzohour & Bühlmann (2016) that shows that a penalized likelihood score can be used to select between different models $p_G^*(y)$ (each fit to an empirical distribution $\hat{p}(y)$), with each assuming a different graph $G \in \mathcal{G}$ (e.g. out of all graphs with up to $d$ nodes). We thus select the graph that has the maximum score:

$$G^* = \arg\max_{G \in \mathcal{G}} \frac{1}{s} \sum_{y \sim \hat{p}(y)}^{s} \ln p_G^*(y) - \frac{|G|}{\ln s}, \tag{7}$$

with $s$ the number of samples, and $|G|$ the number of edges of $G$. This method finds the true causal graph when the causal dependencies in the ANM are all non-linear (Nowzohour & Bühlmann, 2016).

## 4 METHOD

### 4.1 VARIATIONAL INFERENCE WITH A CAUSAL PRIOR

The previous exposition suggests a natural way to simultaneously learn latent variables and fit a causal model: use the ANM (eq. 6) as the prior in the variational optimization problem (eq. 3), by setting $y \equiv z$. This amounts to assuming that the latent variables $z$ have a dependency structure defined by a DAG $G$, and are generated sequentially by application of the non-linear functions $f_i$ corrupted by Gaussian noise $n_i$. Model selection (i.e. finding the optimal graph $G$) can then be done by the score-based search of sec. 3.2.1. Note that this model reduces to a VAE for a DAG with no edges and thus null functions $f_i$:

$$\forall i : \mathrm{pa}(i) = \emptyset \Rightarrow z_i = n_i \text{ (reduction to standard VAE)} \tag{8}$$

At a high level, the same tools used to train a VAE should be applicable in this new setting. However, the ANM is only identifiable (i.e. the true causal directions can be recovered) if the $f_i$ are non-linear (Hoyer et al., 2008), and this makes computing the KL-divergence in eq. 3 intractable, since it is no longer defined in closed form. We will resolve this difficulty by local linearization (sec. 4.4), although the model will still be globally non-linear to ensure identifiability.

One major difference from the exposition by Nowzohour and Bühlmann is that they propose using a non-parametric fitting method over known variables $x$, while we want to simultaneously recover the causal structure and latent variables $z$ (e.g. unknown parameters of objects depicted in images), which must be estimated from inputs $x$ (e.g. raw pixels). Another difference is that their procedure is computationally expensive, as it entails enumerating all graphs $G \in \mathcal{G}$ explicitly. However, given the expressiveness of deep neural network models, in sec. 4.2 we will show how we can replace this search with a simpler model fitting procedure.

### 4.2 MAXIMAL DAG

For the purposes of fitting the model $G$ and $f$ to observed data $x$, we will consider a simplification where we take all ancestors of a variable $z_i$ to be its parents $\mathrm{pa}(i)$, i.e. we replace $G$ with the full DAG with edges $\{i \rightarrow j : i < j, i, j = 1, \ldots, d\}$. The following proposition shows that this will be able to model any underlying SEM, although some of the independence relations will have to be modeled by the learnable edge probabilities described in sec. 4.5.

**Proposition 1.** *The set of all SEMs* $\mathcal{S} = \{(z_i = f_i(z_{pa_G(i)}))_{i=1}^d \mid \forall G, f\}$ *with arbitrary DAGs $G$ and probabilistic functions $f$ is contained within the set* $\mathcal{S}^\Omega = \{(z_i = f_i(z_{1,\ldots,i-1}))_{i=1}^d \mid \forall f\}$, *up to reorderings of the variables $z$.*

*Proof.* See Appendix A. ☐

Prop. 1 says that, as long as the function class of $f$ is expressive enough (as is usually the case with deep networks (Yosinski et al., 2015)), we can find an equivalent SEM using a fixed maximal DAG $G^\Omega$ with edges $\{i \rightarrow j : i < j, i, j = 1, \ldots, d\}$. This simplifies the exposition in the following sections and provides justification for the approximation in sec. 4.5.

### 4.3 Prior probability for linear SEMs

In the simple case when all functions $f_i$ of the ANM are linear, the resulting joint probability must be Gaussian, as it is a linear combination of independent Gaussian noise variables $n_i$ (eq. 5, recalling that $y \equiv z$). However, we need to compute its precise form for general linear ANMs.

**Theorem 2.** *Consider a linear ANM defined as $z_i = a_i^T z_{1,\ldots,i-1} + b_i + n_i$, with $a_i \in \mathbb{R}^{i-1}$, $b_i \in \mathbb{R}$ and $n_i \sim \mathcal{N}(n_i|0, \sigma_i^2)$ for $i = 1, \ldots, d$. Missing edges in the causal graph can be represented as zeros in $a_i$. Then the ANM's joint probability is given by $p(z) = \mathcal{N}(z \mid \mu, \Sigma)$ with*

$$\mu = \left(\prod_{i=d}^{2} A_i\right) b, \quad \Sigma = \left(\prod_{i=d}^{2} A_i\right) \operatorname*{diag}_{i=1,\ldots,d}(\sigma_i^2) \left(\prod_{i=d}^{2} A_i\right)^T, \quad (9)$$

$$A_i = \begin{bmatrix} I_{(i-1)\times(i-1)} & O_{i-1} & O_{(i-1)\times(d-i)} \\ a_i^T & 1 & O_{d-i}^T \\ O_{(d-i)\times(i-1)} & O_{d-i} & I_{(d-i)\times(d-i)} \end{bmatrix}, \quad b = \begin{bmatrix} b_1 \\ \vdots \\ b_d \end{bmatrix},$$

*where $I_{k \times k}$ denotes an identity matrix, while $O_k$ and $O_{k \times l}$ denote a zero column vector and matrix.*

*Proof.* See Appendix C. □

The process in Theorem 2 can be interpreted as a form of mean and covariance propagation: at each stage the ANM applies a linear transformation to the mean and covariance from the previous stage. The matrix $\prod_{i=d}^{2} A_i$ is a lower-triangular matrix representing the edge strengths from parent to child nodes in $G$, obtained from the SEM. This linear transformation is then applied to the SEM biases $b$ to obtain the total mean $\mu$, as well as to the SEM noise variances $\sigma_i^2$ to obtain the total covariance $\Sigma$. For identifiability, however, we must generalize this process to non-linear ANMs, which we do next.

### 4.4 Prior probability for non-linear SEMs

Our approach to deal with non-linear ANMs is to linearize them around a pivot point $z^\circ$. Due to Taylor's theorem this approximation will be accurate for a sufficiently small neighborhood (Nocedal & Wright, 1999, Ch. 1). The ANM's joint probability $p(z)$ will then be Gaussian by Theorem 2.

**Theorem 3.** *The best linear approximation (in the least-squares sense) of an ANM $z_i = f_i(z_{1,\ldots,i-1}) + n_i$ (eq. 5, with missing edges in the causal graph corresponding to ignored inputs in $f_i$), around a pivot point $z^\circ$, is given by eq. 9 (Theorem 2) with*

$$a_i = \left.\frac{\partial f_i(z_{1,\ldots,i-1})}{\partial z_{1,\ldots,i-1}}\right|_{z_{1,\ldots,i-1}=z_{1,\ldots,i-1}^\circ} \quad \text{and} \quad b_i = f_i(z_{1,\ldots,i-1}^\circ) + n_i - a_i^T z_{1,\ldots,i-1}^\circ, \quad (10)$$

*where $z_1^o = n_1$, $z_{1,\ldots,i}^o = f(z_{1,\ldots i-1}) + n_i$ and $n_i \sim \mathcal{N}(0, \sigma_i^2)$ with learnable $\sigma_i$.*

*Proof.* See Appendix D. □

The advantage afforded by Theorem 3 is that the ANM's joint probability $p(z)$ is locally Gaussian, and we can compute its parameters by mean and covariance propagation (eq. 9). Another advantage is that, in the context of training deep networks, one can use automatic differentiation (back-propagation) to compute eq. 10 for arbitrary functions $f_i$, including very expressive ones such as multi-layer perceptrons (MLP). Since this locally-linear SEM is represented as an explicit Gaussian distribution, the objective's KL-divergence (eq. 3) can be computed in closed form (Kingma & Welling, 2014). We can then obtain the full prior distribution by sampling many pivot points $z^o$ by ancestral sampling according to Theorem 3.

### 4.5 Graph search via sparsity

To efficiently search for the graphs in eq. 7, we use Proposition 1 to justify using a fixed maximal graph $G^\Omega$ to learn a *single* SEM $p_f(z)$ (where we make explicit the dependence of $p(z)$ on the learned causal functions $f_i$, c.f. eq. 5). This is in contrast to the previously-proposed technique of enumerating all graphs $G \in \mathcal{G}$ and learning a *different* $p_{f,G}(z)$ for each $G$ (Nowzohour & Bühlmann, 2016). We define the graph of dependencies:

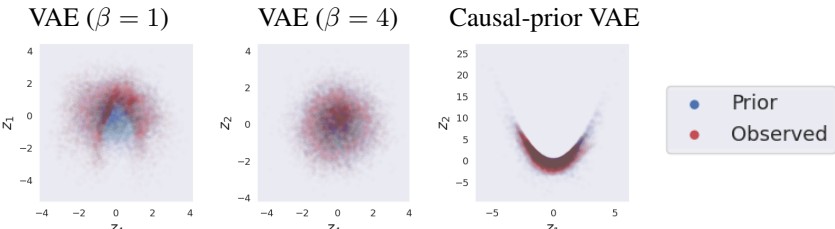

Figure 1: Samples from the observed and prior distributions for different VAEs, showing that the distributions match only for the $\beta = 4$ and the causal-prior VAEs. Only the causal-prior VAE recovers the underlying causal structure that relates the latent variables, a parabola.

**Definition 4.** *We define the **implicit dependency graph** $G(f)$ as the graph with $d$ nodes, and each edge $i \to j$ exists if $f_j(z_{1,\ldots,j-1})$ depends on its $i$th input.*

We can then define the overall objective (from eq. 7 and eq. 3) as

$$p^* = \arg\max_{p \in \mathcal{P},\, q \in \mathcal{Q},\, f} \frac{1}{s} \sum_{\substack{x \sim \hat{p}(z)}}^{s} \left[ \mathbb{E}_{z \sim q(z|x)} \left[ \ln p(x|z) \right] - D_z \left( q(z|x) \| p_f(z) \right) \right] - \frac{|G(f)|}{\ln s}. \qquad (11)$$

To approximate the $|G(f)|$ term that counts the edges of the graph $G(f)$, we introduce edge probabilities $p_{i \to j}$ denoting the probability that an edge from node $i$ to $j$ exists, which allows us to write $|G(f)| = \sum_{j, i < j} p_{i \to j}$ (i.e. for binary probabilities, this recovers the exact edge count). We define each $p_{i \to j}$ as a binary Gumbel-Softmax distribution (Jang et al. (2017), which is suitable for gradient-based optimization. The edge probabilities mask the inputs of the SEM's functions $f_j(z_1, \ldots, z_{j-1})$, as $f_j(p_{1 \to j} z_1, \ldots, p_{j-1 \to j} z_{j-1})$. During model evaluation we can sample binary edge probabilities $p_{i \to j} \in \{0, 1\}$, producing a discrete graph $G$.

We can thus use standard stochastic gradient methods to optimize eq. 11 and obtain an encoder $q$, decoder $p$, and SEM defined by $f$ and $p_{i \to j}$, the latter of which models well-defined causal directions (as long as the underlying functions are non-linear) and reflects independence relations in the data.

### 4.6 Identifiability

In Theorem 7, we demonstrate the identifiability of the proposed model up to some unavoidable indeterminacies. The only transformations that can be implemented by the learned deep networks (encoder, decoder and SEM) and can result in an identical objective value to the optimal model are reorderings of the latent variables, and as well as shifts and orthogonal transformations of the input (implemented by both the encoder at the input, and decoder at the output). The conditions required to achieve this result depend on common features of standard deep networks, namely an encoder and decoder composed of ReLUs and linear operators, batch-normalization in the outputs of the SEM $f$, and a SEM with fixed scale, such as the quadratic used in experiments. The proof is given in Appendix B.

## 5 Experiments

In this section we validate our method on two experiments: the first one learning atemporal variables and the second one learning time-varying variables. In the atemporal experiment (Section 5.1) we show that our method correctly recovers the positions of a shape placed on a parabola given a fixed quadratic prior. In the time-varying experiment (Section 5.2) we show that our method correctly recovers the positions of a moving character and their relationships over time, using a learnable linear prior.

### 5.1 Learning atemporal causal relations

In this section we demonstrate the method's ability to learn causal relationships that are not time dependent. We use a dataset of images of oval shapes at different rotations and scales, placed at

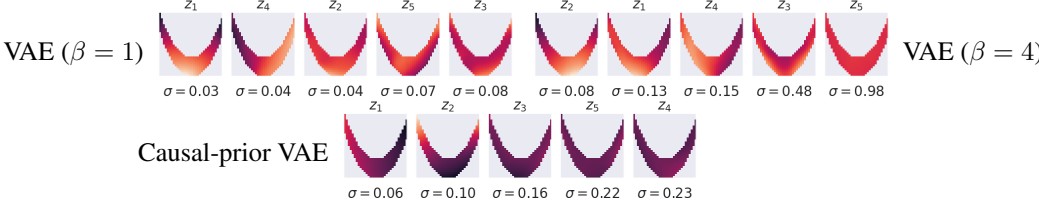

Figure 2: Latent response to the object's position for different VAEs (higher values are brighter). The $\beta = 4$ and $\beta = 1$ VAEs have entangled responses (although it is less entangled for $\beta = 1$), while the causal-prior VAE correctly disentangles the object's horizontal and vertical positions.

positions that follow a parabolic arc $y = x^2 + n$ (with noise $n$). We thus have a causal graph with a single edge $x \to y$. We train our causal-prior VAE with a 3-layer MLP and a parabolic prior (see Appendix E for details), and use the $L_2$ loss between each image and its prediction from the decoder as the reconstruction error (see Appendix **??**). We also train a high-$\beta$ and low-$\beta$ VAEs for comparison (where $\beta$ denotes the multiplicative factor of the KL divergence as in (Higgins et al., 2016a)).

**Latent space visualisation.** In Figure 1 we show the prior (blue) and the predicted distribution (red) for the low-$\beta$ VAE, the high-$\beta$ VAE, and our causal-prior VAE. For the high-$\beta$ VAE (center) the two distributions match, but the latent variables are entangled (i.e. both follow an isotropic Gaussian distribution), and so do not reflect the underlying parabolic relationship between the variables that generate the data ($y = x^2 + n$). For the low-$\beta$ VAE (left), although the observed variables recover the underlying parabolic shape (red), up to a vertical reflection of the coordinates, this distribution does not match the prior at all (blue), since it is constrained to be Gaussian. As such, this generative model $p(x)$ (eq. 2) cannot generate samples that respect the causal relationships in the empirical data distribution. For our causal-prior VAE (right), the two distributions match and both follow the underlying noisy parabolic equation. Our model can thus be used as a generator $p(x)$ that reproduces the true causal structure of the data, a capability that we will explore next in more detail.

**Latent response to position.** Figure 2 shows for different VAEs the value of each latent variable as the position of the shape in the input image is varied, averaged over different rotations and scales. The variables are ordered by their standard deviation ($\sigma$). The goal is to assess how sensitive each variable is to the depicted shape's position. For the causal-prior VAE, the variable $z_1$ clearly corresponds to the shape's $x$ position (with values linearly increasing towards the left), and variable $z_2$ to the $y$ position (with values increasing towards the top), while other variables show minimal response. For the high-$\beta$ VAE the position is entangled across several variables; for the low-$\beta$ VAE the variables $z_4$ and $z_1$ correspond approximately to shape's $x$ and $y$ position, but there is still significant entanglement with the other variables, which are also highly sensitive to the shape's position.

**Latent noise traversal.** In the first and third columns of Figure 3, we show grids of images decoded by each method, obtained by taking one sample and varying the noise $n_i$ of the two latent variables $z_i$ (see eq. 5) that best correspond to horizontal and vertical position. In the first column, decoded images are color-coded by their horizontal position. Since the positions of shapes in this grid are difficult to compare, the second column shows the same images but superimposed, which reveals the underlying parabolic structure of the data. The third column shows the same grid of decoded images, color-coded by vertical position, while the fourth shows them superimposed. While all methods seem to generate images following roughly the parabolic data distribution, we can observe that in both $\beta$-VAE configurations the shapes' positions are entangled (i.e. vary jointly) with their rotation and scale, while the same is not true for the proposed causal-prior VAE. It is interesting to note that in the later case, the noise $n_1$ (corresponding to variable $z_1$) does not map directly to the shape's vertical position (as observed for the $\beta$-VAE), but rather expresses its offset from the parabola, according to $z_1 = z_2^2 + n_1$. In the bottom-right panel, the red tint of images generated from positive noise offsets ($n_1 > 0$) is visible above the parabola, and the blue tint of negative offsets is visible below.

### 5.2 LEARNING TEMPORAL CAUSAL RELATIONS

We now evaluate our method on temporal data, namely short videos. We apply the encoder and decoder independently to each frame, and append an encoded background frame to the the decoder's

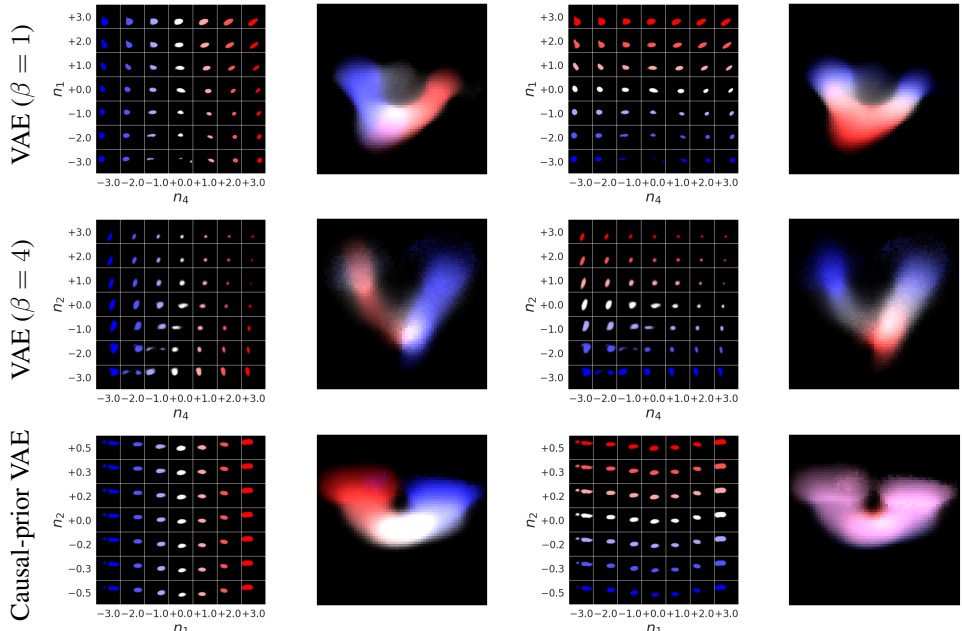

Figure 3: Reconstructed images (as a 2D grid and as superimposed images) as latent noise values are traversed at regular intervals. For the $\beta = 1$ and $\beta = 4$ VAEs the depicted shape's position is entangled with its rotation and scale, while for the causal-prior VAE the horizontal position and vertical offset from the parabola are disentangled correctly.

input (to allow the latent variables to ignore background/time-invariant appearance). The SEM only contains edges in the direction of the arrow of time (see Appendix E for details). We create a dataset based on the Super Mario Bros video game depicting an object moving with linear motion in different directions, at different speeds, and on varied backgrounds. Three frames are observed, with the object's coordinates $(x_i, y_i)$ for $i \in \{1, 2\}$ sampled uniformly from $[7, 12]$, and the third obeying $x_3 = 2x_2 - x_1, y_3 = 2y_2 - y_1$. We train our method with a linear prior, and otherwise similarly to Sec. 5.1. (see Appendix F for details) and select the sparsest graph that still achieves the best reconstruction accuracy. We use the reconstruction error given by $\sum_{t=1}^{3} L_2(x_t, \hat{x}_t) + L_2(x_3, \hat{x}_3^f)$ where $x_t$ denotes a frame at time $t$, $\hat{x}_t$ denotes a prediction made using the decoder $p(\hat{x}_t|z_t)$ and $\hat{x}_3^f$ denotes a prediction made using the decoder $p(\hat{x}_3^f|f_3(z_1, z_2))$ (see Appendix **??** for details). For quantitative evaluation of learned graphs please refer to Appendix G.

**Learning a causal graph.** Figure 4 shows the graph learned by the causal-prior VAE, with the bottom row $(z_1, z_2, z_3, z_4)$ corresponding to latent variables at time $t = 1$, the middle row $(z_5, z_6, z_7, z_8)$ corresponding to $t = 2$ and the top row $(z_9, z_{10}, z_{11}, z_{12})$ corresponding to $t = 3$. Learned edges are shown in black and missing ones in beige; learned functions $f_i$ are represented as a number (multiplicative factor) next to the corresponding edge, since they are linear in this case. The graph shows that each component of the object's 2D position at time $t = 3$ (expressed as $(z_{10}, z_{11})$) depends on its 2D positions at times $t = 1$ $(z_2, z_3)$ and $t = 2$ $(z_6, z_7)$, following the model $z_{10} = f_{10}(z) = 2.3z_6 - 1.1z_2$ and $z_{11} = f_{11}(z) = 2.2z_7 - 1.1z_3$, which matches the data generation process up to a scale factor.

**Latent response to position.** Figure 5 shows the response of each latent variable as the object's horizontal and vertical position is varied on the input. The rows correspond to variables from times $t = 3, 2, 1$ respectively. We can see that $z_2, z_6, z_{10}$ encode the object's position along the bottom-right to top-left diagonal at different moments in time, while $z_3, z_7, z_{11}$ encode the position along the orthogonal direction. The remaining variables do not exhibit any significant response to position. This shows the variables correctly match the data generation process up to a rotation.

**Interventions visualisation.** Once a SEM relating the latent variables has been learned, we can perform interventions on some of its variables. In Figure 6 we encode a randomly sampled reference

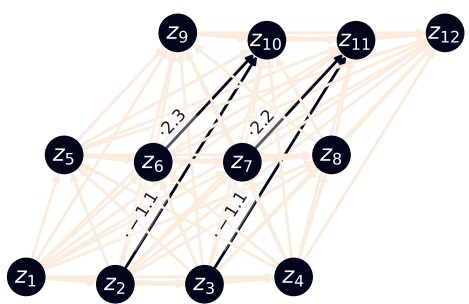

Figure 4: Learned graph edges and mechanisms relating the latent variables, recovering the data generation process up to a scaling factor.

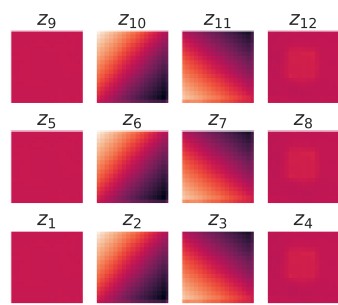

Figure 5: Latent response of variables to the object's horizontal and vertical position, recovering the data generation process up to a rotation.

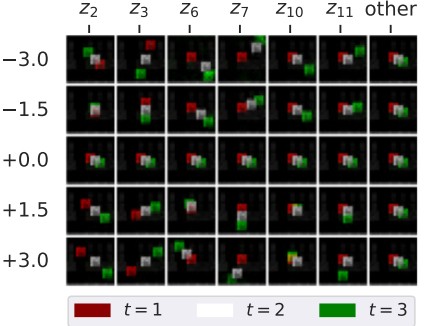

Figure 6: Intervention results showing that intervening on each variable propagates to its descendants through the correct mechanism.

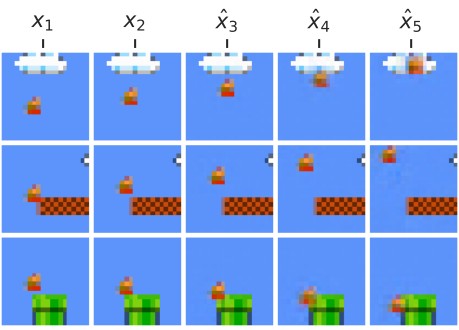

Figure 7: Extrapolation results obtained by iteratively passing the encoded variables through the learned SEM to predict their future values.

clip from the dataset and separately intervene by assigning different values (rows) to variables $z_1, ..., z_{12}$ (columns), propagating the changes to their child variables using the SEM, and decoding the results. The results are decoded using a black background and the resulting 3-frame decoded sample is combined into a single frame by colouring the $t = 1$ decoded image by red, the $t = 2$ image by white, and $t = 3$ by green. We observe that intervening on the variables for $t = 1$ ($z_2, z_3$) changes both the object's position in $t = 1$ (red, columns 1,2) and $t = 3$ (green, columns 1,2), following the SEM. Intervening on the variables for $t = 2$ ($z_6, z_7$) changes the object's $t = 2$ position (white, columns 3,4) and $t = 3$ position (green, columns 3,4); while intervening on the variables for $t = 3$ ($z_{10}, z_{11}$) changes only its $t = 3$ position (green, columns 5,6), which reflects the fact that other positions (in the past) do not depend on it. Intervening on other variables has no effect (last column). This confirms that the model has learned the correct temporal causal structure, as each intervention affects the right variables.

**Extrapolation visualisation.** Having learned a SEM relating those at time $t$ to the variables at times $(t - 1, t - 2)$ now makes it possible to extrapolate the variable values into the future. In Figure 7 we start with 3 samples (rows) of 2 consecutive frames from the dataset $x_1, x_2$, encode them to obtain the $t = 1, 2$ latent variables ($z_1, ..., z_8$), and iteratively pass them through the SEM to compute the $t = 3$ latent variables ($z_9, ..., z_{12}$), decoding these into $\hat{x}_3$. We now repeat this process by taking $z_5, ..., z_{12}$ as the $t = 1, 2$ variables and use the SEM to predict the $t = 3$ variables which we decode into $\hat{x}_4$ and similarly for $\hat{x}_5$, thus obtaining predictions for 3 time steps into the future. The future predictions confirm the knowledge learned in the SEM that the object moves linearly and can be used to predict future frames accurately.

## 6 CONCLUSIONS

In this work we proposed a general model that naturally extends the variational learning framework to learn a non-linear Structural Equation Model (SEM) as the prior, which enables causal learning to be performed on perceptual modalities such as images and video. To reconcile the non-linearity of SEMs while using Gaussian variables we proposed a fully differentiable method that locally linearises the SEM to obtain a locally-Gaussian distribution. Furthermore, we relaxed the search over causal graphs as a joint continuous optimization over non-linear causal functions $f$. The proposed method shows promise to scale to high-dimensional data and moderately complex SEMs, however future work should explore more large-scale data such as long videos and other input modalities.

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

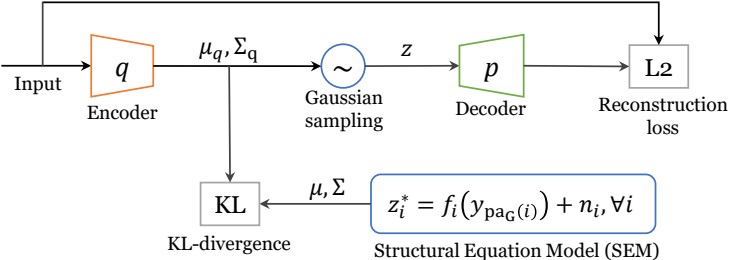

Figure 8: Overview of the causal-prior VAE architecture.

# APPENDIX

## A  PROOF OF DAG MAXIMALITY

**Proposition 5.** *The set of all SEMs $\mathcal{S} = \{(z_i = f_i(z_{pa_G(i)}))_{i=1}^d \,|\, \forall G, f\}$ with arbitrary DAGs $G$ and probabilistic functions $f$ is contained within the set $\mathcal{S}^\Omega = \{(z_i = f_i(z_{1,...,i-1}))_{i=1}^d \,|\, \forall f\}$, up to reorderings of the variables $z$.*

*Proof.* Because $G$ is a DAG it has an acyclic ordering of its vertices (Bang-Jensen & Gutin, 2009); i.e. if $G$ has vertices $V = \{v_1, ..., v_n\}$ we can always construct a sequence $(z_1, ..., z_n)$ such that the set $Z = \{z_1, ..., z_n\}$ is mapped one-to-one to the set $V$, and such that $z_i$ is an ancestor of $z_j$ (i.e. there exists a path from $z_i$ to $z_j$ in $G$) for all $i < j$. Therefore, the set of parents $\mathrm{pa}_G(z_j)$ has to be a subset of the set $\{z_i | i < j\}$, from which Proposition 5 follows. □

## B  JOINT IDENTIFIABILITY OF REPRESENTATION AND CAUSAL MECHANISM

Before proving our identifiability result, we must first introduce a lemma about the commutativity of certain piece-wise linear functions.

**Lemma 6.** *Consider a Leaky ReLU activation function (He et al., 2015), defined as:*

$$\mathcal{R}_{C,D}(x) = \begin{cases} Cx & \text{if } x \geq 0 \\ Dx & \text{if } x < 0 \end{cases}, \tag{12}$$

*with constants $C > 0, C \neq 1, D > 0, D \neq 1, C \neq D$. Then the class of functions $\phi$ that commute with R, i.e.*

$$\phi(\mathcal{R}_{C,D}(x)) = \mathcal{R}_{C,D}(\phi(x)), \forall x \tag{13}$$

*is the set of monotonic piece-wise linear homogeneous functions with one piece in each half-plane, i.e.*

$$\phi(x) = \begin{cases} Ax & \text{if } x \geq 0 \\ Bx & \text{if } x < 0 \end{cases}, \quad A, B > 0. \tag{14}$$

*Proof.* To solve the commutativity identity in Eq. 13, we partition it into four domains, namely $\{x \geq 0, x < 0\} \times \{\phi(x) \geq 0, \phi(x) < 0\}$:

$$\text{If } x < 0, \phi(x) < 0 \quad \Rightarrow \quad C\phi(x) = \phi(Cx) \quad \Rightarrow \quad \phi(x) = Ax \tag{15}$$
$$\text{If } x < 0, \phi(x) \geq 0 \quad \Rightarrow \quad C\phi(x) = \phi(Dx) \quad \Rightarrow \quad \phi(x) = 0 \tag{16}$$
$$\text{If } x \geq 0, \phi(x) < 0 \quad \Rightarrow \quad D\phi(x) = \phi(Cx) \quad \Rightarrow \quad \phi(x) = 0 \tag{17}$$
$$\text{If } x \geq 0, \phi(x) \geq 0 \quad \Rightarrow \quad D\phi(x) = \phi(Dx) \quad \Rightarrow \quad \phi(x) = Bx \tag{18}$$

Equations 15 and 18 amount to solving the equality $Ef(x) = \phi(Ex)$ which is satisfied for any homogeneous linear function $\phi$ assuming $E \neq 1$. Equations 16 and 17 amount to solving the equality $Ef(x) = \phi(Fx)$ which is only satisfied for $\phi(x) = 0$ assuming $E \neq 1, E \neq F$. Combining the results from Equations 15-18 the result follows. □

We can now introduce our main result.

**Theorem 7.** *For a causal VAE model (Eq. 3 and Eq. 5 with $y \equiv z$), assume the following conditions:*

1. *The encoder $q(z|x)$ and decoder $p(x|z)$ are Gaussians (Eq. 4) parameterized by deep networks, containing a Leaky ReLU (Eq. 12) as its last layer and first layer, respectively.*

2. *The outputs of the SEM $f_i$ are de-meaned and normalized, i.e. they are composed with batch normalization (BN) operators: $z_i = \mathrm{BN}(f(z_{\mathrm{pa}(i)}))$ where $\mathrm{BN}(u) = (u - \mathbb{E}[u])/\sqrt{\mathrm{Var}[u]}$, or they have a fixed constant scale.*

*Then the model is identifiable up to the following indeterminacies, i.e. denoting the optimal parameters by $\theta^*$, there is a different set of parameters $\theta$ such that $p_{\theta*}(x) = p_\theta(x)$ only under the following learnable transformations:*

1. *Simultaneous shifts by $b$ and orthogonal transformations $R$ of the encoder's input and decoder's output, i.e.: $\mu_q(x) \leftarrow \mu_q(Rx + b)$, $\mu_p(z) \leftarrow R^{-1}\mu_p(z) - b$.*

2. *Latent variable permutations, i.e. reorderings of $z$.*

3. *If the SEM mechanisms $f_i$ contain additional symmetries (e.g. $f_i = f_i \circ S$ for some operator $S$), then there will be indeterminacy up to application of that operator.*

*Proof.* The training objective is to find the model parameters $\theta$ of the generative distribution $p_\theta(x)$ such that it matches the observed empirical distribution $p(x)$. The parameters are assumed to exist if the function class of $p_\theta(x)$ is sufficiently expressive, as is the case in over-parameterized deep networks. We also assume that there exists a true model $\theta^*$ which generates the data distribution $p_{\theta*}(x)$. Then the claim of non-identifiability is that there exists at least one different parameterization $\theta$ for which $p_\theta(x) = p_{\theta*}(x)$, i.e. it is possible to learn another model that also fits the true data distribution but has different parameters $\theta$. We can express the lower bound on the data distribution using the evidence lower bound (ELBO, Eq. 3)

$$\ln p_{\theta*}(x) \geq \mathbb{E}_{x \sim p(x)}[\mathbb{E}_{q_\theta(z|x)}[\ln p_\theta(x|z)] - \mathrm{KL}[q_\theta(z|x)||p_\theta(z)]] \tag{19}$$

with encoder $q_\theta(z|x)$, decoder $p_\theta(x|z)$, and learnable prior $p_\theta(z)$ generated using the SEM (slightly overloading $\theta$ to include all parameters of the model).

Under non-identifiability, we assume that we can find another set of parameters $\theta$ that fit the data perfectly: $p_\theta(x) = p_{\theta*}(x)$. In that case Eq. 19 becomes a strict equality, implying the following two conditions:

$$\ln p_{\theta*}(x) = \mathbb{E}_{x \sim p(x)}[\mathbb{E}_{q_\theta(z|x)}[\ln p_\theta(x|z)]] \tag{20}$$

$$0 = \mathbb{E}_{x \sim p(x)}[\mathrm{KL}[q_\theta(z|x)||p_\theta(z)]], \tag{21}$$

In words, the expectation of the log likelihood (first term of right-hand side of Eq. 19) becomes the exact log evidence $\ln p_{\theta*}(x)$, while the posterior $q_\theta(z|x)$ fits the prior $p_\theta(z)$ exactly (their KL-divergence is zero). Furthermore, for the optimal model $\theta^*$ we know that

$$\ln p_{\theta*}(x) = \mathbb{E}_{x \sim p(x)}[\mathbb{E}_{q_{\theta*}(z|x)}[\ln p_{\theta*}(x|z)]] \tag{22}$$

Note the subtle difference from Eq. 20 is to use $\theta^*$ instead of $\theta$. We can thus combine the results for both models $\theta^*$ (Equation 22) and $\theta$ (Equation 20) as

$$\mathbb{E}_{x \sim p(x)}[\mathbb{E}_{q_\theta(z|x)}[\ln p_\theta(x|z)]] = \mathbb{E}_{x \sim p(x)}[\mathbb{E}_{q_{\theta*}(z|x)}[\ln p_{\theta*}(x|z)]] \tag{23}$$

We will write the left-hand side (LHS) of Eq. 23 in a form that makes its equivalence classes more obvious, using function composition. In order to do that, we first recall Eq. 4, which we rewrite here (making $\theta$ explicit) for convenience:

$$q_\theta(z|x) = \mathcal{N}(z|\mu_\theta(x), \Sigma_\theta(x)) \tag{24}$$

The reparametrization trick, which is at the core of VAE implementations, allows us to sample from the probabilistic encoder $q_\theta(z|x)$ with a deterministic function $q_{\theta,\epsilon}(x)$ (typically a deep network) that takes samples $\epsilon$ from a fixed normal distribution and applies an affine transformation to them:

$$q_{\theta,\epsilon}(x) = \mu_\theta(x) + \Sigma_\theta(x)\,\epsilon, \quad \epsilon \sim \mathcal{N}(0,1). \tag{25}$$

The inner expectation in the LHS of Eq. 23 can then be written as:

$$\mathbb{E}_{z \sim q_\theta(z|x)}[\ln p_\theta(x|z)] = \mathbb{E}_{\epsilon \sim \mathcal{N}(0,1)}[\ln p_\theta(x|q_{\theta,\epsilon}(x))]. \tag{26}$$

By Eq. 4 the decoder $p_\theta(x|z)$ is also a Gaussian with parameterized mean $\mu_\theta^p(z)$,[1] so Eq. 23 is equivalent to:

$$\mathbb{E}_{x\sim p(x)}\mathbb{E}_{z\sim q_\theta(z|x)}\ln p_\theta(x|z) = \mathbb{E}_{x\sim p(x)}\mathbb{E}_{z\sim q_\theta(z|x)}\left[-\gamma\left\|\mu_\theta^p(z)-x\right\|^2\right] \tag{27}$$

$$= \mathbb{E}_{x\sim p(x)}\mathbb{E}_{z\sim q_\theta(z|x)}\left[L_x(\mu_\theta^p(z))\right], \quad L_x(x') = -\gamma\left\|x'-x\right\|^2, \tag{28}$$

absorbing constants into $\gamma$. Finally, the LHS of Eq. 23 is then equivalent to the expectation of a composition of deterministic functions (applied from right to left):

$$\mathbb{E}_{x\sim p(x),\epsilon\sim p(\epsilon)}[L_x \circ \mu_\theta^p \circ q_{\theta,\epsilon} \circ x]. \tag{29}$$

This makes explicit the order of operations in computing the reconstruction loss, and separates out all non-deterministic elements into the expectation (namely by using the reparameterization trick with $\epsilon \sim p(\epsilon) = \mathcal{N}(0,1)$).

We can now enumerate the equivalence classes of Eq. 23, by inserting identity operators $I = g^{-1}\circ g = h^{-1}\circ h$ (for arbitrary invertible functions $g, h$) between compositions of learnable functions in Eq. 29. Inserting these identities, Eq. 29 is equivalent to

$$\mathbb{E}_{x\sim p(x),\epsilon\sim p(\epsilon)}[L_x \circ (h\circ h^{-1})\circ \mu_\theta^p \circ (g^{-1}\circ g)\circ q_{\theta,\epsilon}\circ(h\circ h^{-1})\circ x] \tag{30}$$

$$= \mathbb{E}_{x\sim p(x),\epsilon\sim p(\epsilon)}[(L_x\circ h)\circ \underbrace{(h^{-1}\circ \mu_\theta^p\circ g^{-1})}_{\bar{\mu}_\theta^p}\circ \underbrace{(g\circ q_{\theta,\epsilon}\circ h)}_{\bar{q}_{\theta,\epsilon}}\circ \underbrace{(h^{-1}\circ x)}_{\bar{x}}] \tag{31}$$

where we grouped the arbitrary functions with the decoder as $\bar{\mu}_\theta^p$, with the encoder as $\bar{q}_{\theta,\epsilon}$, and with the input as $\bar{x}$. We can then apply the substitution $\bar{x} = h^{-1}\circ x$, using the fact that

$$L_x(x') = \gamma\left\|h(x')-h(x)\right\|^2 = L_{h\circ x}(h\circ x') \tag{32}$$

is only ever true if $h$ is an orthogonal linear transformation plus a constant (since Euclidean distances are invariant only under generalized rotations and translations),[2] and thus obtaining the equivalent reparameterization

$$\mathbb{E}_{\bar{x}\sim p(\bar{x}),\epsilon\sim p(\epsilon)}[L_{\bar{x}}\circ \bar{\mu}_\theta^p\circ \bar{q}_{\theta,\epsilon}\circ \bar{x}]. \tag{33}$$

Thus the function class of $h$ must necessarily be restricted to orthogonal transformations, induced by the Euclidean structure of $L$ (Eq. 32).

As for $g$, it is restricted by the fact that it must be absorbed into the encoder, i.e. in Eq. 31 we must be able to group it with the previous encoder $q_{\theta,\epsilon}$ and define a new encoder

$$\bar{q}_{\theta,\epsilon} = g\circ q_{\theta,\epsilon}\circ h \tag{34}$$

that can still be implemented as a deep network (of the same function class as $q_{\theta,\epsilon}$). Since we require that the encoder be followed by a Leaky ReLU $\mathcal{R}_{C,D}$ (assumption 1 of Theorem 7), for this to be true, $g$ must commute with $\mathcal{R}_{C,D}$, i.e.:

$$\bar{q}_{\theta,\epsilon} = g\circ q_{\theta,\epsilon}\circ h = g\circ \underbrace{\mathcal{R}_{C,D}\circ q'_{\theta,\epsilon}}_{q_{\theta,\epsilon}}\circ h = \mathcal{R}_{C,D}\circ g\circ q'_{\theta,\epsilon}\circ h \tag{35}$$

where the first equality is taken from Eq. 31, in the second equality we decompose the network $q_{\theta,\epsilon}$ into its final Leaky ReLU $\mathcal{R}_{C,D}$ and the rest of the network $q'_{\theta,\epsilon}$, and finally in the last step we use the commutativity of $g$ and $\mathcal{R}_{C,D}$ (Lemma. 6). Since by Lemma. 6 only monotonic piece-wise linear homogeneous functions commute with $\mathcal{R}_{C,D}$, this restricts the class of admissible functions for $g$ to that class.

An identical conclusion follows for $g^{-1}$, as long as the first layer of the decoder is also a Leaky ReLU (by assumption 1).

---

[1]Assuming identity covariance for simplicity, a common assumption in implementations, and which does not materially change the result.

[2]This is the same reason why the function $h$ and its inverse are used twice in Eq. 30 instead of inserting two different functions, e.g. $h_1$ and $h_2$ and their inverses; the same generalized rotation $h$ must be applied to both inputs of the Euclidean distance in Eq. 32.

This means that the latent variables are non-identifiable up to $g$ implementing individual rescaling of each variable and variable permutation, unless we impose further constraints. We can impose mild structural constraints on the SEM to fix the rescaling, namely assuming that the output of each $f_{\theta,i}(z_{\text{pa}(i)})$ has fixed scale. This can be achieved by either: 1) batch-normalization; 2) a SEM with no learnable scale. Batch-normalization, which is used by many common deep network models, fixes the distribution mean to zero and the variance to one for each dimension, removing scaling and shifting degrees-of-freedom. As a special case, the quadratic model used in Section 5.1) does not have learnable scale parameters. The only remaining degrees of freedom are variable permutations, which are unavoidable in latent variable models.

Having characterized the equivalence classes of Eq. 20, we must now consider those of Eq. 21. Using the same reparameterization trick as in Eq. 25, the KL-divergence in Eq. 21 is equivalent to

$$\text{KL}[q_\theta(z|x)||p_\theta(z)] = \mathbb{E}_{z \sim q_\theta(z|x)}[\ln q_\theta(z|x) - \ln p_\theta(z)] \tag{36}$$

$$= \mathbb{E}_{\epsilon \sim p(\epsilon)}[\ln q_{\theta,\epsilon}(x) - \ln p_\theta(q_{\theta,\epsilon}(x))] \tag{37}$$

Therefore, we can also express it using a composition of operators:

$$\mathbb{E}_{x \sim p(x), \epsilon \sim p(\epsilon)}[L \circ q_{\theta,\epsilon} \circ x - L \circ \mu_\theta^f \circ q_{\theta,\epsilon} \circ x], \tag{38}$$

with $\mu_\theta^f$ the mean of the Gaussian output by $f$ (by Theorem 2), assuming variance one as before for simplicity. We can now insert the same identity operators and follow an identical derivation to Eq. 30-33, which recovers the exact same equivalence classes as before. The only identity that can be added differently from Eq. 30 will be an operator $S$ for which $\mu_\theta^f$ is invariant ($\mu_\theta^f = \mu_\theta^f \circ S$), so indeterminacy up to such symmetries of $f$ is the only other possibility.

## C   PROOF OF MEAN AND VARIANCE PROPAGATION FOR LINEAR ANMS

**Theorem 8.** *Consider a linear ANM defined as $z_i = a_i^T z_{1,\dots,i-1} + b_i + n_i$, with $a_i \in \mathbb{R}^{i-1}$, $b_i \in \mathbb{R}$ and $n_i \sim \mathcal{N}(n_i|0, \sigma_i^2)$ for $i = 1, \dots, d$. Missing edges in the causal graph can be represented as zeros in $a_i$. Then the ANM's joint probability is given by $p(z) = \mathcal{N}(z \,|\, \mu, \Sigma)$ with*

$$\mu = \left(\prod_{i=d}^{2} A_i\right) b, \quad \Sigma = \left(\prod_{i=d}^{2} A_i\right) \operatorname*{diag}_{i=1,\dots,d}(\sigma_i^2) \left(\prod_{i=d}^{2} A_i\right)^T, \tag{39}$$

$$A_i = \begin{bmatrix} I_{(i-1)\times(i-1)} & O_{i-1} & O_{(i-1)\times(d-i)} \\ a_i^T & 1 & O_{d-i}^T \\ O_{(d-i)\times(i-1)} & O_{d-i} & I_{(d-i)\times(d-i)} \end{bmatrix}, \quad b = \begin{bmatrix} b_1 \\ \vdots \\ b_d \end{bmatrix},$$

*where $I_{k \times k}$ denotes an identity matrix, while $O_k$ and $O_{k \times l}$ denote a zero column vector and a zero matrix, respectively.*

*Proof.* We can write the ANM $z_i = a_i^T z_{1,\dots,i-1} + b_i + n_i$ in a recursive multivariable form as

$$z_{1,\dots,i} = \begin{bmatrix} I_{(i-1)\times(i-1)} \\ a_i^T \end{bmatrix} z_{1,\dots,i-1} + \begin{bmatrix} O_{i-1} \\ b_i \end{bmatrix} + \begin{bmatrix} O_{i-1} \\ 1 \end{bmatrix} n_i \tag{40}$$

Applying the formula for linear combination of Gaussians (Bishop, 2006, Ch. 8.1.4)

$$x \sim \mathcal{N}(x|\mu_x, \Sigma_x), y \sim \mathcal{N}(y|\mu_y, \Sigma_y) \implies Ax + By + c \sim \mathcal{N}(x|A\mu_x + B\mu_y + c, A\Sigma_x A^T + B\Sigma_y B^T) \tag{41}$$

to eq. 40 we can express the mean and covariance of $z_{1,\dots,i}$ as a function of the mean and covariance for $z_{1,\dots,i-1}$ as

$$\mu_{1,\dots,i} = \begin{bmatrix} I_{(i-1)\times(i-1)} \\ a_i^T \end{bmatrix} \mu_{1,\dots,i-1} + \begin{bmatrix} O_{i-1} \\ b_i \end{bmatrix} \tag{42}$$

$$\Sigma_{1,\dots,i} = \begin{bmatrix} I_{(i-1)\times(i-1)} \\ a_i^T \end{bmatrix} \Sigma_{1,\dots,i-1} \begin{bmatrix} I_{(i-1)\times(i-1)} \\ a_i^T \end{bmatrix}^T + \begin{bmatrix} O_{i-1} \\ 1 \end{bmatrix} \sigma_i^2 \begin{bmatrix} O_{i-1} \\ 1 \end{bmatrix}^T$$

where we have used

$$\mu_{1,\dots,i} = \mathbb{E}\left[z_{1,\dots,i}\right], \quad \Sigma_{1,\dots,i} = \mathbb{E}\left[(z_{1,\dots,i} - \mu_{1,\dots,i})(z_{1,\dots,i} - \mu_{1,\dots,i})^T\right], \tag{43}$$

and for the noise variables $n$ it holds that $\mathbb{E}\left[n\right] = 0$ and $\mathbb{E}\left[nn^T\right] = \mathrm{diag}_i(\sigma_i^2)$. Now assume that eq. 39 holds for some $d = k$. Inserting expressions for $\mu_{1,\dots,k}$ and $\Sigma_{1,\dots,k}$ from eq. 39 into eq. 42 and extending the matrices with zeros and ones we obtain

$$\mu_{1,\dots,k+1} = \begin{bmatrix} I_{k\times k} & O_k \\ a_{k+1}^T & 1 \end{bmatrix} \left(\prod_{i=k}^{2}\begin{bmatrix} A_{i,d=k} & O_k \\ O_k^T & 1 \end{bmatrix}\right)\begin{bmatrix} b_{1,\dots,k} \\ 0 \end{bmatrix} + \begin{bmatrix} O_k \\ b_{k+1} \end{bmatrix} \tag{44}$$

$$\Sigma_{1,\dots,k+1} = \begin{bmatrix} I_{k\times k} & O_k \\ a_{k+1}^T & 1 \end{bmatrix} \left(\prod_{i=k}^{2}\begin{bmatrix} A_{i,d=k} & O_k \\ O_k^T & 1 \end{bmatrix}\right)\begin{bmatrix} \mathrm{diag}_{i=1,\dots,k}(\sigma_i^2) & O_k \\ O_k^T & 0 \end{bmatrix}$$

$$\left(\prod_{i=k}^{2}\begin{bmatrix} A_{i,d=k} & O_k \\ O_k^T & 1 \end{bmatrix}\right)^T \begin{bmatrix} I_{k\times k} & O_k \\ a_{k+1}^T & 1 \end{bmatrix}^T + \begin{bmatrix} O_{k\times k} & O_k \\ O_k^T & 1 \end{bmatrix}\sigma_{k+1}^2$$

Identifying

$$A_{k+1,d=k+1} = \begin{bmatrix} I_{k\times k} & O_k \\ a_{k+1}^T & 1 \end{bmatrix}, \quad A_{i,d=k+1} = \begin{bmatrix} A_{i,d=k} & O_k \\ O_k^T & 1 \end{bmatrix}, \quad b_{1,\dots,k+1} = \begin{bmatrix} b_{1,\dots,k} \\ b_{k+1} \end{bmatrix} \tag{45}$$

eq. 44 becomes

$$\mu_{1,\dots,k+1} = \left(\prod_{i=k+1}^{2} A_{i,d=k+1}\right)b_{1,\dots,k+1} \tag{46}$$

$$\Sigma_{1,\dots,k+1} = \left(\prod_{i=k+1}^{2} A_{i,d=k+1}\right)\mathrm{diag}_{i=1,\dots,k+1}(\sigma_i^2)\left(\prod_{i=k+1}^{2} A_{i,d=k+1}\right)^T \tag{47}$$

which completes the inductive step $d = k \rightarrow k + 1$. Finally, we apply eq. 41 to relation 40 for $d = 2$ to obtain

$$\mu_{1,2} = \begin{bmatrix} 1 \\ a_2 \end{bmatrix}\mu_1 + \begin{bmatrix} 0 \\ b_2 \end{bmatrix} = A_2 b_{1,2} \tag{48}$$

$$\Sigma_{1,2} = \begin{bmatrix} 1 \\ a_2 \end{bmatrix}\Sigma_1\begin{bmatrix} 1 \\ a_2 \end{bmatrix}^T + \begin{bmatrix} 0 \\ 1 \end{bmatrix}\sigma_2^2\begin{bmatrix} 0 \\ 1 \end{bmatrix}^T = A_2\mathrm{diag}(\sigma_1^2, \sigma_2^2)A_2^T$$

where we have used $\mu_1 = b_1$ and $\Sigma_1 = \sigma_1^2$, which shows that eq. 39 holds for $d = 2$. Therefore, by the induction principle the relation 39 holds for all $d \geq 2$. $\square$

## D    PROOF OF LINEARISATION OF NON-LINEAR SEMs

**Theorem 9.** *The best linear approximation (in the least-squares sense) of an ANM $z_i = f_i(z_{1,\dots,i-1}) + n_i$ (eq. 5, with missing edges in the causal graph corresponding to ignored inputs in $f_i$), around a pivot point $z^\circ$, is given by eq. 39 (Theorem 8) with*

$$a_i = \left.\frac{\partial f_i(z_{1,\dots,i-1})}{\partial z_{1,\dots,i-1}}\right|_{z_{1,\dots,i-1}=z^\circ_{1,\dots,i-1}} \quad \text{and} \quad b_i = f_i(z^\circ_{1,\dots,i-1}) + n_i - a_i^T z^\circ_{1,\dots,i-1}, \tag{49}$$

*where $z^\circ_{1,\dots,i-1} = f(z_{1,\dots i-1}) + n_i$ and $n_i \sim \mathcal{N}(0, \sigma_i^2)$ with learnable $\sigma_i$.*

*Proof.* By Taylor's theorem, expanding $f_i(z_{1,\dots,i-1}) + n_i$ around $z^\circ_{1,\dots,i-1}$ up to first order gives

$$f_i(z_{1,\dots,i-1}) \approx f_i(z^\circ_{1,\dots,i-1}) + n_i + (z_{1,\dots,i-1} - z^\circ_{1,\dots,i-1})^T \left.\frac{\partial f_i(z_{1,\dots,i-1})}{\partial z_{1,\dots,i-1}}\right|_{z_{1,\dots,i-1}=z^\circ_{1,\dots,i-1}} \tag{50}$$

$$\approx a_i^T z_{1,\dots,i-1} + b_i \tag{51}$$

where $a_i$ and $b_i$ are given by eq. 49. We can now use this linearisation of $f_i$ to define a linearised SEM as $z_i = a_i^T z_{1,\dots,i-1} + b_i$ and using this with Theorem 8 the result follows. $\square$

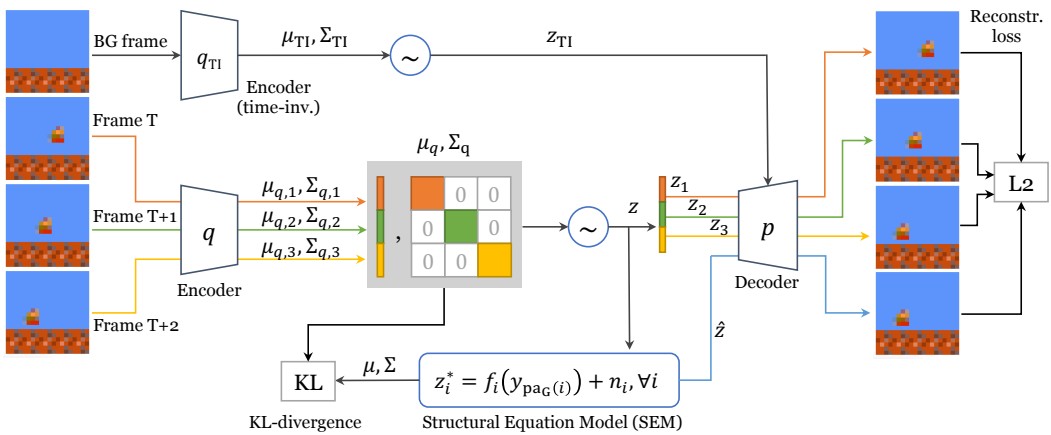

Figure 9: The temporal causal-prior VAE architecture, for a sequence of 3 video frames (plus a random frame to represent time-invariant information, such as background).

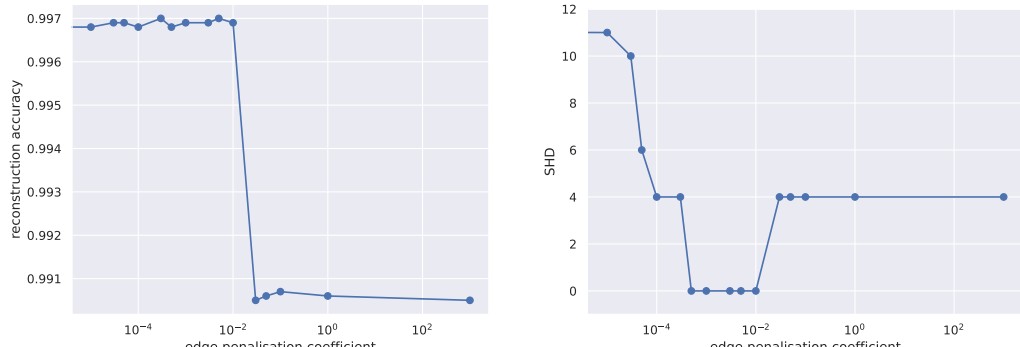

Figure 10: Reconstruction accuracy as a function of parameter controlling graph sparsity.

Figure 11: Structural Hamming Distance as a function of parameter controlling graph sparsity.

## E    TRAINING DETAILS FOR THE CAUSAL-PRIOR VAE

We used a custom dataset of 76800 binary images of size $64 \times 64$ containing ovals generated at 6 different scales $s$, 40 rotations $r$, 32 horizontal positions $x$ and 10 vertical positions $y$, where $y = x^2 + n_y$ and $s$ and $r$ factors are sampled independently. We trained our causal-prior VAE and two isotropic-prior VAEs for comparison, one with a lower $\beta$ and one with higher $\beta$ (where $\beta$ denotes the factor multiplying the KL divergence as defined in (Higgins et al., 2016a)). The encoder is a 3-layer MLP with 64 hidden ReLU units in each layer; the decoder is identical but with 4 layers, and there are 5 latent variables $z$. We use the Adagrad optimizer (Duchi et al., 2011) with learning rate 0.003 on batches of 100 samples, until convergence. The architecture is shown in Figure 8.

## F    TRAINING DETAILS FOR THE TEMPORAL CAUSAL-PRIOR VAE

We created a custom dataset of 3-frame $20 \times 20$ px video sequences of the main Super Mario Bros character moving linearly in a random direction and with a random speed on different backgrounds, where the character's positions are given by $x_1, y_1, x_2, y_2 \sim \mathcal{U}(7, 12)$, $x_3 = 2x_2 - x_1$, $y_3 = 2y_2 - y_1$ where $x_i$ and $y_i$ are the horizontal and vertical position at frame $i$ and $\mathcal{U}$ is the uniform distribution. We train our causal-prior VAE with 12 variables, 4 per time frame, and allow the SEM to learn arbitrary linear relationships between them. The architecture is shown in Figure 9. Each dataset sample consists of a tuple consisting of a background and 3 consecutive frames where the character moves linearly. Each of the 3 frames is then encoded separately into a 4-variate Gaussian distribution

and these are concatenated to form a 12-variate Gaussian, which is compared with its closest match in the SEM-sampled local Gaussian distribution. The distribution is then sampled and split into 4 samples per time frame and these are concatenated with the background latents and decoded back into 3 frames which are then compared using $L_2$ loss with the 3 frames on the input. Additionally, the latents from the first two frames are passed through the SEM to predict their value at the next time frame and this is decoded and compared using $L_2$ loss with the third frame on the input. The architecture is shown in Figure 9.

## G  QUANTITATIVE EVALUATION OF LEARNED TEMPORAL GRAPHS

**Reconstruction accuracy vs. graph complexity**  Figure 10 shows for the temporal experiments the reconstruction accuracy achieved with the model as a function of the parameter controlling graph sparsity ($-1/\ln(s)$ in Equation 7) after training for 250 epochs. The plot shows that for weak edge penalisation the reconstruction accuracy is good (around 99.7%) while if the edge penalisation is too large the accuracy drops (to around 99.1%). This is because for weak edge penalisation the graph is relatively dense which allows the SEM to model the time-based causal relationships between the variables and when the penalisation is too big the graph becomes too sparse to be able to model these relationships. Somewhere around the value of the coefficient $10^{-2}$ the graph becomes as sparse as possible while still keeping the reconstruction accuracy high, and this is the area from which we select the graph.

**Structural Hamming Distance vs. graph complexity**  Figure 11 shows for the temporal experiments the Structural Hamming Distance between the learned and the ground truth graph as a function of the parameter controlling graph sparsity ($-1/\ln(s)$ in Equation 7) after training for 250 epochs. The SHD is computed by counting how many edges need to be inserted or removed to obtain the ground truth graph (up to a permutation of variables within each time step). In the range where the coefficient is below approx. $10^{-3}$ the edge penalisation is too weak resulting in a graph with too many edges (thus a high SHD) and in the region above approx. $10^{-2}$ the graph becomes too sparse with no edges (thus also resulting in high SHD). The region between $10^{-3}$ and $10^{-2}$ corresponds to the region where the learned graph has exactly the same structure as the ground truth graph (and thus SHD is zero).

