# OpenReview forum: "Unsupervised Learning of Causal Relationships from Unstructured Data"
_ICLR.cc/2023/Conference — Submitted to ICLR 2023_

### Official Review · Reviewer_8bM1 · 2022-10-19

**Confidence:** 4
**Correctness:** 4
**Technical Novelty And Significance:** 4
**Empirical Novelty And Significance:** 4
**Recommendation:** 8

**Clarity, Quality, Novelty And Reproducibility:**

The paper is clearly written, with a concise overview of the methods,  experimental setup, and results.  It builds on existing VAEs, but is distinct and appears novel.  The experiments are on simple data sets, but indicate an ability to discover causal relationships.


**Strength And Weaknesses:**

Strengths:
- Novel approach to modeling, building on well-established techniques like VAEs.
- Experimental results support the authors' claims.
- Well-written paper.

Weaknesses:
- would have been nice to see experiments on more complex data sets


**Summary Of The Paper:**

The authors propose a VAE-like framework with SEM to represent causal relationships among latent variables, and train the model by linearization.  Experiments on images and video demonstrate an ability to discover basic causal relationships.


**Summary Of The Review:**

I enjoyed reading this paper, and found the exposition clear and results interesting.  I see it as a solid contribution.

Question: In Figure 4, why do z6 and z7 not depend on z2 and z3?

---

> ### Author Response · Authors · 2022-11-14
> **Response**
>
> > It would have been nice to see experiments on more complex data sets.
>
> We thank the reviewer for the suggestion. We have considered using more complex datasets, however they pose an additional set of difficulties such as being more difficult to evaluate (often there is no ground truth causal graph and mechanisms are not known), and require focusing on the design of more specialised architectures. In this work we aimed to present a proof of concept that works on image data, which is high-dimensional compared to more structured data (e.g. tabular data). We plan to focus on more complex data sets in future work.
>
> > Question: In Figure 4, why do z6 and z7 not depend on z2 and z3?
>
> $z_6$ and $z_7$ are independent of $z_2$ and $z_3$ because of the way we generate the data -- we sample $x_{t=1},y_{t=1},x_{t=2},y_{t=2}$ independently (which correspond to $z_2$, $z_3$, $z_6$, $z_7$) and then compute $x_{t=3}=2x_{t=2}-x_{t=1}$ and $y_{t=3}=2y_{t=2}-y_{t=1}$ (which correspond to $z_{10}$, $z_{11}$), so the positions at the first two time frames are independent while the positions at the third frame depend on the positions at the first two frames. Please refer to Appendix E for more details.

---

### Official Review · Reviewer_qqHC · 2022-10-23

**Confidence:** 3
**Correctness:** 3
**Technical Novelty And Significance:** 2
**Empirical Novelty And Significance:** 2
**Recommendation:** 3

**Clarity, Quality, Novelty And Reproducibility:**

The paper is rather clear, although I have been confused by the discussion of identifiability, and the possibly inadequate claims.

Minor comments:

+ The theoretical concept of maximal DAG (in Prop 1), is perfectly acceptable, but is essentially equivalent to the idea that the set of DAGs can be reparameterized by the set of permutations (GSP, IGSP, and many algorithms used this idea before).



**Strength And Weaknesses:**

Strength:

(1) The problem that the authors are interested in is important.

(2) The maximal dag idea allows to circumvent a lot of the cumbersome machinery in causal discovery learning for acyclicity, and search for topological ordering

(3) The idea of linearization is interesting (I had not seen it in causal inference before)


Weaknesses:

(1) There is a discussion of identifiability in the introduction regarding the structural equation model for the latent variables (supposedly treating them as observed). Then, there is another discussion of identifiability in the setting of decoder + SEM setting, pointing out some previous work. However, to the best of my understanding, the 2008 paper is concerned with observed random variables. Because here the zs are latent, I do not think the theory actually holds. If it does, please explain it in details in the paper, as it is not straightforward. Also, the setting with temporality is another instance, and identifiability could hold in one setting, but not the other.

(2) A discussion of time complexity is missing. In particular, what is the complexity of putting together the covariance matrix detailed in equation (8), and how does it affect scaling with respect to the number of latent variables? It may look prohibitively expensive for large dimensions.

(3) There is no discussion of whether the algorithm has any form of theoretical guarantees after using a lower bound, and after linearization.

(4) The experimental section seems rather like a discussion of disentanglement capabilities, rather than causal discovery learning. Why was 5 latent variables used in the first experiment, and not 2? After all, there are only one cause and one effect latent variable? Similarly, if the paper goes through all the process of explaining how to fit VAEs with latent SEMs, I would expect experiments with evaluation of the graph quality (e.g., does the method discover a sparse set of interactions between latent variables, is the conditional distribution exactly parabolic).

(5) I could not find quantitative evaluation of methods in the main text. If the authors claim the VAE is identifiable, then could they report MCC and other metrics imported from non-linear ICA? I would also expect the authors to report metrics over the learned causal graphs for example.

**Summary Of The Paper:**

This paper is concerned with learning causal relationship between latent variables of a deep generative model, used to model high-dimensional complex data such as images. This is an important and difficult topic in causal inference. The paper starts with background about VAEs, SEMs and then proceeds to describing their method. The proposed method consists in linearizing the SEM given by a "maximal DAG", in which the topological ordering is fixed, and each latent variable is densely connected to each of the ones appearing before it in the ordering. For linear SEMs, the likelihood function is directly computable. Then, two experiments on image datasets are presented, in which certain variables have causal relationships (e.g., position of sprites in images), and one would like the DGMs to learn those relationships.

**Summary Of The Review:**

I have found this paper to be concerned with an important problem, and proposing an interesting method. However, I think that the paper neither has solid theoretical discussion of the setting or of the accuracy of the method, and also only has weak empirical evaluations. Therefore, I am leaning towards rejection. I am looking forward to the discussion period w/ authors, as well as reviewers.

---

> ### Author Response · Authors · 2022-11-14
> **Response**
>
> > (1) Because here the zs are latent, I do not think the theory actually holds. If it does, please explain it in details in the paper, as it is not straightforward.
>
> We thank the reviewer for pointing this out – the theoretical results mentioned in Section 4.6 indeed are for observed variables, and so only guarantee identifiability of a SEM when the encoder/decoder are fixed, not for the general case. This requires a slightly more involved theoretical argument, which we will post as a general response to all reviewers.
>
> > (2) What is the complexity of putting together the covariance matrix detailed in equation (8), and how does it affect scaling with respect to the number of latent variables?
>
> The time complexity in putting together the covariance matrix in Equation 8 is linear in the number of latent variables. This is because in order to compute the values for $a_i$ and $b_i$ in Equation 8 we need to evaluate Equation 9 for i from 1 to N, where N is the number of latent variables. This comes from the fact that $a_i$ and $b_i$ depend on the linearisation points $z_{1,...,i-1}^o$ (with i=2,...,N) which need to be sampled ancestrally from the graph with N nodes, which is a linear time complexity operation.
>
> > (3) There is no discussion of whether the algorithm has any form of theoretical guarantees after using a lower bound, and after linearization.
>
> For theoretical guarantees regarding identifiability, please see answer to (1). For theoretical guarantees regarding the evidence lower bound, we show this derivation step by step throughout Sections 3 and 4, resulting in Equation 10.
>
> > (4) Why was 5 latent variables used in the first experiment, and not 2? After all, there are only one cause and one effect latent variable?
>
> We decided to use 5 latent variables instead of 2 to demonstrate that our method is robust to having unused variables. This is because, in practice, we often do not know the minimum number of latent variables that are sufficient for describing a particular dataset, so by setting it to a sufficiently large number we can still recover the solution. This is also done in other works such as the beta-VAE (Higgins et al., 2016a) which uses 10 variables, even though only 5 of them express known factors in the data.
>
> > Similarly, if the paper goes through all the process of explaining how to fit VAEs with latent SEMs, I would expect experiments with evaluation of the graph quality (e.g., does the method discover a sparse set of interactions between latent variables, is the conditional distribution exactly parabolic).
> >
> >
> > If the authors claim the VAE is identifiable, then could they report MCC and other metrics imported from non-linear ICA?
>
> It can be seen in Figure 4 that the learned graph corresponds exactly to the ground truth graph; a direct comparison to the ground truth (and why the learned graph is correct) is elaborated on in Section 5.2, paragraph “Learning a causal graph”.
> We have also added quantitative evaluation results in Appendix F, showing Reconstruction Accuracy and Structural Hamming Distance metrics plotted against model complexity. We are in the process of calculating additional metrics, including Mean Correlation Coefficient, that we will also add before the end of the discussion period. This will include the Log-Likelihood of the parabolic fit.

---

> > ### Comment · Reviewer_qqHC · 2022-11-19
> > **Acknowledging response**
> >
> > I would like to thank the authors for responding to my questions. I have looked over the new theory, and it seems to me that it is quite hastily written (e.g., typos, imprecisions) and confusing. Several points were raised by the other reviewer. I will add to this that identifiability is a property of the model, and doesn't involve the encoder. In particular, a different statement should be written for the inference procedure.

---

> > > ### Author Response · Authors · 2022-11-22
> > > **Response**
> > >
> > > >I would like to thank the authors for responding to my questions. I have looked over the new theory, and it seems to me that it is quite hastily written (e.g., typos, imprecisions) and confusing.
> > >
> > > We thank the reviewer for their feedback. While we will strive to correct any typos or imprecisions in the camera ready version, it would be helpful if the reviewer could point to any specific examples in the proof that they find confusing, imprecise or incorrect and we will also explain these in the comment.
> > >
> > > >I will add to this that identifiability is a property of the model, and doesn’t involve the encoder. In particular, a different statement should be written for the inference procedure.
> > >
> > > In causal representation learning, both the model of causal relationships (SEM) and the model of the variables (the encoder/decoder) can co-adapt to create indeterminacies. A simple example of this is variable permutation: the encoder/decoder can predict variables in a different order, and the SEM is permuted in a consistent way. So studying identifiability must be done jointly for both at the same time, since they conspire together to create indeterminacies that would not exist with just one in isolation. This is what the proof is focused on.

---

> > > > ### Comment · Reviewer_qqHC · 2022-11-22
> > > > **Response**
> > > >
> > > > Sure. I must apologize, but due to the limited amount of time I have, I will point out only one mistake.
> > > >
> > > > With all due respect, I think the transition from (19) to (20) and (21) makes no sense. If the ELBO is tight, then you have equality between the left hand side and the right hand side. This tell you nothing about the term in the left-side of (21) to be zero! What is zero is the variational gap (i.e., KL(q(z | x) || p(z | x)) = 0, for every data point x). It seems to be that the authors confused the variational gap KL(q(z | x) || p(z | x) for the KL(q(z |x) || p(z)) term present in the VAE loss function.
> > > >
> > > > Given this mistake, I cannot consider the rest of the proof. I encourage the authors to work again on this proof and submit to the next conference.

---

### Official Review · Reviewer_mc4c · 2022-10-24

**Confidence:** 4
**Correctness:** 1
**Technical Novelty And Significance:** 2
**Empirical Novelty And Significance:** 2
**Recommendation:** 1

**Clarity, Quality, Novelty And Reproducibility:**

Quality: In my humble opinion, the paper suffers from a fatal flaw – I think causal representations cannot identifiable from unlabelled observational data alone, at least not without additional assumptions. If this is correct, then unfortunately I believe that there is no reason for the proposed method to work.

Novelty: There are already several papers that propose causal representation learning with a VAE with a causally structured prior. The key novelty of the paper is the claimed identifiability from unsupervised observational data. If that result is correct and well supported, then that is certainly sufficient novelty (and potentially very impactful).

Clarity: The paper is clearly written.


**Strength And Weaknesses:**

Strengths:
- The problem of causal representation learning is interesting and potentially very impactful.
- The data regime that the authors tackle – unsupervised observational data – is the holy grail of causal representation learning: a strong identifiability result here could be immediately practically relevant.
- The VAE ansatz with a fully connected graph is very sensible.
- The paper is generally well written. I appreciate the thorough introductions to the background material.

Weaknesses:
- Unfortunately, I believe that the main premise of the paper is flawed. (I hope I am wrong and the authors can correct me!) The authors correctly point out that when the causal variables are given, nonlinear ANMs are identifiable. They then claim that this implies that *causal representations with a latent nonlinear ANM are identifiable*, i.e. one can identify both the map from low-level data to causal variables and the causal graph from data. As far as I understand it, they do this without making assumptions on the decoder (like that being linear). I am afraid this claim of identifiability is not correct.
  - One known counterexample is the special case of the trivial graph. Locatello et al ["Challenging common assumptions in the unsupervised learning of disentangled representations", ICML 2019] and some other papers show that in this case the variables are not identifiable.
  - Another counterexample can be constructed from any nonlinear ANM with causal variables $z_i$ and noise variables $\epsilon_i$. We can define a second nonlinear ANM that has a trivial graph and causal variables $z'_i = \epsilon_i$ such that the observational data distribution remains the same. (Of course, the map to the data space will then be different between the two causal models.)
  - If I'm missing something here, I kindly ask the authors to add an identifiability argument (for the representations, not just the causal structure once the representations are known) to the paper.
- I don't quite understand the reasoning behind the linearizing approximation. Why not just use a nonlinear density estimator like a normalizing flow for each conditional probability density in the prior? Then we can approximate the KL quite well by sampling. Unlike the linearization, this approach should be unbiased (in some limit) and is well established.
- I find it difficult to draw conclusions from the experiments. It would be better if the results were analyzed more quantitatively, showing that the learned representations and graphs are correct.
- In the related work section, several recent works that use VAEs with causal structure in the latent space are missed, for instance:
  - von Kugelgen et al, "Self-Supervised Learning with Data Augmentations Provably Isolates Content from Style", NeurIPS 2021
  - Brehmer et al, "Weakly supervised causal representation learning"
  - Ahuja et al, "Interventional Causal Representation Learning"
  - as a review, it may also be useful to point to Schölkopf et al, "Towards Causal Representation Learning", IEEE

**Summary Of The Paper:**

The paper considers the problem of causal representation learning from observational data without any form of supervision. The authors assume that the latent causal structure follows a nonlinear additive noise model (ANM) and claim that it is then identifiable. In practice, they use a VAE with a causally structured prior. One novelty in this is that they propose to approximate the KL term in the ELBO by linearizing the causal mechanisms. Their approach is demonstrated on toy image data as well as time series data from a video game.

**Summary Of The Review:**

If the author's claim is true, this would be a great result. Unfortunately, I do not think that the main point – that causal representations can be identified from observational data if we just assume the latent causal structure to be a nonlinear ANM – is correct. I hope the authors can point me to what I'm missing, add an identifiability argument, or provide strong empirical evidence for this claim. Otherwise, I do not think that this paper can be accepted.

UPDATE after rebuttal and discussion: I am grateful to the authors for constructively and quickly engaging with my questions and criticisms and even working out a new identifiability claim. Unfortunately, I am convinced that this claim is incorrect, but even if it were true, it would not solve the main issue with the paper. I am afraid that this paper will need more substantial work before acceptance, and recommend rejection at ICLR.

---

> ### Author Response · Authors · 2022-11-14
> **Response**
>
> > Unfortunately, I believe that the main premise of the paper is flawed.
> > If I'm missing something here, I kindly ask the authors to add an identifiability argument (for the representations, not just the causal structure once the representations are known) to the paper.
>
> We thank the reviewer for pointing this out -- it is indeed an important distinction, which warrants a non-trivial extension of our theoretical support. Similarly to previous papers, we have to make some assumptions about the SEM and encoder/decoder (which are respected in our experiments), to narrow down the equivalence class of solutions enough that we can obtain identifiability “up to” some unavoidable symmetries (such as variable reordering). We are preparing a detailed derivation that we will post as a general comment, and we hope that it will satisfy the reviewer (and if not, we will try to improve it before the discussion ends).
>
> > Why not just use a nonlinear density estimator like a normalizing flow for each conditional probability density in the prior?
>
> The reviewer is correct in pointing out that there are different density estimators which we could use, and there are many valid solutions to this problem. However, using normalizing flows as the density estimator requires the functions to be invertible, but causal models are not invertible in general. For example, if we want to use a quadratic relationship as in Section 5.1 (which is non-invertible), this would not be possible with normalizing flows.
>
> > It would be better if the results were analyzed more quantitatively, showing that the learned representations and graphs are correct.
>
> We have added quantitative evaluations in Appendix F, showing the Reconstruction Accuracy and the Structural Hamming Distance. Both metrics are plotted for varying amounts of model complexity. We are in the process of calculating additional metrics, including Mean Correlation Coefficient, that we will also add before the end of the discussion period.
>
> > In the related work section, several recent works that use VAEs with causal structure in the latent space are missed, for instance:
>
> We thank the reviewer for the suggestions and will add them to the related work section of the paper.

---

> > ### Comment · Reviewer_mc4c · 2022-11-16
> > **Thanks for the response**
> >
> > Thanks to the authors for the response and for engaging with my questions and criticisms. I am looking forward to (and very curious about) the new theory. I will update my review after seeing that.
> >
> > For now, I just want to clarify one point:
> >
> > > However, using normalizing flows as the density estimator requires the functions to be invertible, but causal models are not invertible in general. For example, if we want to use a quadratic relationship as in Section 5.1 (which is non-invertible), this would not be possible with normalizing flows.
> >
> > What I had in mind were conditional normalizing flows in which a bijective map from noise to a given causal variable is conditional on the causal parents. Then there is no requirement of the parent dependency being invertible.

---

> > ### Comment · Reviewer_mc4c · 2022-11-18
> > **Identifiability claim has substantial issues**
> >
> > I have responded to the top-level post with the new identifiability theorem. I believe there are several issues with it.
> >
> > Overall, I thank the authors for their hard work on an important problem. Unfortunately, I remain convinced that the main premise of the paper is fundamentally flawed. In my opinion, it cannot be accepted at ICLR.
> >
> > I hope that the authors can fix these issues and can turn this into a great paper at a future venue.

---

### Official Review · Reviewer_WspD · 2022-10-25

**Confidence:** 5
**Correctness:** 2
**Technical Novelty And Significance:** 1
**Empirical Novelty And Significance:** 2
**Recommendation:** 3

**Clarity, Quality, Novelty And Reproducibility:**

The paper is clearly written. There are major issues with the quality of the work, please refer to the weaknesses section. While the paper is original but the lack of proper description of the method, poor experimentation makes it very weak.

**Strength And Weaknesses:**

**Strengths**
The authors study an important and timely problem. Currently, we lack methods that can recover latents with learnable causal priors.
The authors have done a good job of exposition starting with basics of the VAE and building the structure of the paper.

**Weaknesses**

I have many concerns with the paper that I highlight in a bulleted form below.

1. **Method for local linearization** The authors state that they make a piecewise linear approximation of the non-linear SEM around pivot point. The authors do not state how they select the pivot points. Since the approximation is valid in a neighborhood only, it is important to have sufficiently many pivot points. This is central to the paper and it is appalling that it is missing. Further, if there was such a method used a natural question to ask would be if the method has some identification guarantees or it is completely ad-hoc.



2. **Experiments have major issues** There are two sets of experiments that authors carry out. In the first experiment the authors use a quadratic relationship between the latents.

   In the first set of experiments the authors use the known quadratic prior. If the prior is not learned then the experiment is a mere sanity check and does not provide any valuable insight. In fact, if the model is already aware that the relationship is quadratic, then the SEM effectively becomes linear by treating z^2 as the parent feature. If the SEM becomes linear, then doesn't that go against the whole point of the paper.

   In the second set of experiments, the authors resort to using a linear SEM again. I am not sure why authors do that. If the point of the paper is to achieve identification for non-linear SEMs why use linear SEMs. Also, if it is linear and Gaussian would you not run into non-identification issues.

2.  **Crucial comparisons are missing** The authors operate in time series settings to conduct their main experiments. For these experiments, why do they not compare against the work https://proceedings.mlr.press/v177/lachapelle22a/lachapelle22a.pdf. In the work referenced the authors learn a causal graph in time. See Section 4, Figure 3 of the paper. The authors rely on time-sparsity as the regularization to achieve identification.

3.  **Lack of theoretical guarantees**  As highlighted above, the paper does not provide valuable experimental insights so I wanted to comment on theoretical guarantees. The paper also does not provide theoretical guarantees either. For instance, many of the works that authors cite do provide identification guarantees. The current theorems in the paper do not provide any new insights.

4. I would recommend the authors to do a significant revamp of the experiments. Also, it should be made clear how you select pivots and manage the non-linear case. These changes will improve the paper. Finally, any insights from the theory even for two variable case would be valuable.

5. The authors in Section 4.6 for some reason state the identification guarantees for non-linear SEM are empirically shown. There are many works and this https://arxiv.org/pdf/1205.2536.pdf is one example where identification has been shown in multivariate case.


**Summary Of The Paper:**

In traditional causal inference, one often assumes access to the structured variables and the task is to discover causal relationships between them. In recent years, there has been a growing amount of interest to tackle the question when these variables themselves are unknown and this task is termed as "causal representation learning". To this end, the first task is to infer these ``causal variables" themselves. Some existing works such as the causal VAE have tackled this question but in the context of linear structural equation models for the latents. In this work, the authors extend it to general non-linear SEMs. The authors leverage additive models as they can be identified with observational data. The method proposed by the authors extends standard VAEs that leverage independence of latents as prior to a non-linear SEM driven prior. To train VAEs, it is far more efficient to have a closed form expression for the KL divergences, which one cannot with standard non-linear SEM based prior. To tackle this the authors propose to break SEM into a piecewise linear SEM. The authors validate their method on synthetic datasets.

**Summary Of The Review:**

The current work proposes a new class of VAEs that accommondate a SEM based prior. The authors propose a variational approximation for it. However, the method used for arriving at the approximation is not described, central parts of it namely pivot selection is missing. The main contribution of the paper is supposed to be non-linear additive SEMs but the main experiments are conducted with linear SEMs. Key comparisons with other works are missing as well.

---

> ### Author Response · Authors · 2022-11-14
> **Response**
>
> > *Method for local linearization.* The authors do not state how they select the pivot points. Since the approximation is valid in a neighborhood only, it is important to have sufficiently many pivot points.
>
> This implementation detail was indeed missing – we thank the reviewer for noticing it. The pivot points $z^o$ in Equation 9 are selected by ancestrally sampling the graph, with learnable variance for the noise variables. The reviewer correctly states that it is important to have sufficiently many pivot points, which we do by drawing 100 samples at each optimisation step. We will include this description in the paper.
>
> > *Experiments have major issues.* In fact, if the model is already aware that the relationship is quadratic, then the SEM effectively becomes linear by treating z^2 as the parent feature. If the SEM becomes linear, then doesn't that go against the whole point of the paper.
>
> The reviewer states that using a fixed quadratic prior is equivalent to using a linear SEM. We must point out that this cannot be true, as a linear SEM would not be able to recover variables whose relationship is quadratic (visualized in Figure 1, right). Further, because $z_1$ is Gaussian, $z_2=z_1^2$ is no longer Gaussian and so cannot be modeled using only a linear Gaussian model, instead requiring a non-linear treatment.
>
> > In the second set of experiments, the authors resort to using a linear SEM again. I am not sure why authors do that.
>
> We use a linear SEM in this experiment because it is the simplest setting in which we can demonstrate our method with time-based data. Dealing with temporal data introduced additional necessary complexities to the problem already, so we wanted to minimize the experiment’s complexity in other ways. There is no reason why it shouldn’t work for a non-linear model, as we demonstrated in the experiments without time-based data (Section 5.1).
>
> > If the point of the paper is to achieve identification for non-linear SEMs why use linear SEMs. Also, if it is linear and Gaussian would you not run into non-identification issues.
>
> Because the variables are conditioned on time, the SEM with known variables is still identifiable even if we use linear relationships and Gaussian variables, as long as there are no instantaneous effects (Peters et al. 2013).
>
> > *Crucial comparisons are missing.* For these experiments, why do they not compare against the work https://proceedings.mlr.press/v177/lachapelle22a/lachapelle22a.pdf.
>
> We thank the reviewer for the suggestion. As this is a work concurrent with ours, we could not do a detailed experimental comparison, but we will cite it and discuss it.
>
> > As highlighted above, the paper does not provide valuable experimental insights
> > I would recommend the authors to do a significant revamp of the experiments.
>
> We would like to kindly ask the reviewer to give specific criticisms of the experiments, as it is hard to counter a generic assessment. If specific suggestions are made, we will strive to implement them.
>
> > *Lack of theoretical guarantees.* For instance, many of the works that authors cite do provide identification guarantees. The current theorems in the paper do not provide any new insights.
> > Finally, any insights from the theory even for two variable case would be valuable.
>
> We provide new insights on aspects other than identifiability, such as a novel linearisation method for efficient construction of local Gaussian approximations to arbitrary non-linear ANMs. However, to strengthen these results, we will soon present a theoretical argument on identifiability as a general response to all reviewers, that we hope the reviewer will consider adequate as a new theoretical insight.
>
> > The authors in Section 4.6 for some reason state the identification guarantees for non-linear SEM are empirically shown. There are many works and this https://arxiv.org/pdf/1205.2536.pdf is one example where identification has been shown in multivariate case.
>
> We agree with the reviewer that the cited proof by Hoyer et al. (2008) is only limited to the two-variable case, while the multivariable case is only shown experimentally – we will refer to the suggested reference instead, as it provides a stronger result, and thank the reviewer for mentioning it.

---

### Author Response · Authors · 2022-11-17
**Response about Identifiability**

In Theorem 6 (revised manuscript), we demonstrate the identifiability of the proposed model up to some unavoidable indeterminacies.
The only transformations that can be implemented by the learned deep networks (encoder, decoder and SEM) and can result in an identical objective value to the optimal model are reorderings of the latent variables, as well as shifts and orthogonal transformations of the input (implemented by both the encoder at the input, and decoder at the output).
The conditions required to achieve this result depend on common features of standard deep networks, namely an encoder and decoder composed of ReLUs and linear operators, batch-normalization in the outputs of the SEM $f$, and either ReLUs in the SEM or a non-rotationally-symmetric SEM, such as the quadratic used in experiments.
In case the true SEM includes symmetries, there will of course also be a corresponding ambiguity in the solution, which we account for in the theorem.
The proof is given in Appendix B.

---

> ### Comment · Reviewer_mc4c · 2022-11-18
> **Severe doubts about this claim**
>
> Thanks to the authors for addressing my / our questions by this substantial extension of the original theory, especially in such a short time.
>
> However, there are several aspects of this theorem that I do not understand. In no particular order:
> 1. I believe that assumption 1 is vacuous and does not actually impose any restriction on the function class of these networks. For instance, f(x) = ReLU(x) - ReLU(-x) = x is the identity.
> 2. In assumption 3 there is a typo: $f(z) \neq Rf(z) \forall R \in O(n)$ does not make sense, I assume what is meant here is $\exists R \in O(n): f(z) \neq f(Rz)$?
> 3. Anyway, what exactly is the function $f(z)$? I have only seen the definition for individual functions $f_i$. Sorry if I missed this.
> 4. Transformation 2 is a special case of transformation 1.
> 5. I am not sure if I understand the statement in transformation 3. Can you make this more precise?
> 6. Since the theorem only claims identifiability up to orthogonal transformations among the causal variables, does it make sense to talk about identifying the causal graph at all? Say the true causal variables are $z_1$ and $z_2$ with some true causal graph between them. Now we learn the causal variables are $z'_1 = (z_1 + z_2) / \sqrt{2}$ and $z'_2 = (z_1 + z_2) / \sqrt{2}$. Which graph would then be correct? The theorem makes no statement about this, but was this not a main goal of the paper?
>
> All in all, I appreciate the effort the authors have put into this, but I have strong doubts about this theorem.

---

> > ### Author Response · Authors · 2022-11-19
> > **Response to doubts**
> >
> > We would like to thank the reviewer for giving feedback and helping us improve our proof, which we greatly appreciate. On each point:
> >
> > 1. It is true that requiring the encoder to end with a ReLU+linear layer, i.e. have the form $\phi(x)=b+W \max(0, \phi’(x))$, is not restrictive enough and can become vacuous with the suggested construction. The motivation behind this constraint was to limit co-adaptation between the encoder and decoder to only affine relations, leveraging the non-invertibility of ReLUs. However, we updated assumption 1 to instead require $\phi(x)=\mathcal{R}(\phi’(x))$, with $\mathcal{R}$ a Leaky ReLU operator. Without a learnable linearity after this operator, it is no longer possible to implement such a construction that makes the assumption vacuous, since only $\phi’(x)$ is free to be chosen. Furthermore, the Leaky ReLU has the attractive property that it only commutes with a very narrow subset of all functions, which we prove in Lemma 6, and this allowed us to remove assumption 3.
> >
> > 2. This is indeed a typo, we meant to write $\nexists R \in \{O(n) \setminus \{I\} \}: p_f(z) = p_f(Rz)$. However, we removed assumption 3, given that assumption 1 is now more restrictive.
> >
> > 3. Indeed, $f(z)$ is the function obtained by concatenating all $f_i(z)$. We will instead refer to ${f_i(z)}$.
> >
> > 4. Although a permutation is a special case of an orthogonal transformation, we note that transformation 2 (i.e. an orthogonal transformation) is applied to the encoder’s input and the decoder’s output, whereas transformation 1 (i.e. a permutation) is applied to the latent variables.
> >
> > 5. What we are claiming here is that we can recover the mechanisms $f$ only up to the symmetries inherently present in them. For example, if the data is generated according to $z_2=f_2(z_1)=z_1^2$ we can only determine $z_1$ up to the indeterminacies of this quadratic function, i.e. up to bilateral symmetry $S(z_1)=-z_1$.
> >
> > 6. We only claim identifiability up to orthogonal transformations between the encoder’s input and decoder’s output (consequence 1), not for the latent variables. For the latent variables we claim that they are identifiable up to permutations (consequence 2).

---

### Decision · Program_Chairs · 2023-01-20

**Decision:**

Reject

**Justification For Why Not Higher Score:**

Reviewers believe that the paper suffers from a fatal flaw

**Justification For Why Not Lower Score:**

The considered problem is interesting and important.

**Metareview: Summary, Strengths And Weaknesses:**

This paper aims to learn causal relationships from purely observational unstructured data. The problem is interesting. Unfortunately, reviewers believe that the paper suffers from a fatal flaw--the nonlinear additive noise model can find causal structure over measured variables, but there is no guarantee that it can help find latent variables as well.